# The AT_1_/AT_2_ Receptor Equilibrium Is a Cornerstone of the Regulation of the Renin Angiotensin System beyond the Cardiovascular System

**DOI:** 10.3390/molecules28145481

**Published:** 2023-07-18

**Authors:** Mélissa Colin, Céline Delaitre, Sébastien Foulquier, François Dupuis

**Affiliations:** 1CITHEFOR, Université de Lorraine, F-54000 Nancy, France; melissa.colin@univ-lorraine.fr (M.C.); celine.delaitre@univ-lorraine.fr (C.D.); 2Department of Pharmacology and Toxicology, MHeNS—School for Mental Health and Neuroscience, Maastricht University, 6200 MD Maastricht, The Netherlands; s.foulquier@maastrichtuniversity.nl; 3CARIM—School for Cardiovascular Diseases, Maastricht University, 6200 MD Maastricht, The Netherlands

**Keywords:** AT_1_ receptor, AT_2_ receptor, angiotensin II, AT_1_/AT_2_ balance, biased agonism, TRV120027, *S*-nitrosation

## Abstract

The AT_1_ receptor has mainly been associated with the pathological effects of the renin-angiotensin system (RAS) (e.g., hypertension, heart and kidney diseases), and constitutes a major therapeutic target. In contrast, the AT_2_ receptor is presented as the protective arm of this RAS, and its targeting via specific agonists is mainly used to counteract the effects of the AT_1_ receptor. The discovery of a local RAS has highlighted the importance of the balance between AT_1_/AT_2_ receptors at the tissue level. Disruption of this balance is suggested to be detrimental. The fine tuning of this balance is not limited to the regulation of the level of expression of these two receptors. Other mechanisms still largely unexplored, such as *S*-nitrosation of the AT_1_ receptor, homo- and heterodimerization, and the use of AT_1_ receptor-biased agonists, may significantly contribute to and/or interfere with the settings of this AT_1_/AT_2_ equilibrium. This review will detail, through several examples (the brain, wound healing, and the cellular cycle), the importance of the functional balance between AT_1_ and AT_2_ receptors, and how new molecular pharmacological approaches may act on its regulation to open up new therapeutic perspectives.

## 1. Introduction

Over the past decades, the team of Professor Jeffrey Atkinson, who recently passed away (1943–2023) and who taught Pharmacology at the Faculty of Pharmacy of Nancy for over 25 years, has contributed to demonstrating the major role of the renin angiotensin system (RAS) in the regulation of the cardiovascular system and the cerebral circulation [1,2,3,4].

The RAS is an important hormonal system involved in numerous physiological processes, as shown by the copious literature devoted to the exploration and comprehension of this system since the discovery of renin in the late 19th century by Robert Tigerstedt [5]. The systemic RAS is known for its involvement in vascular homeostasis, blood pressure regulation, and sodium and water retention in the kidney [6]. Angiotensinogen, a glycoprotein produced by the liver and released in the blood is the first element of the systemic RAS. When blood pressure drops, the juxtaglomerular apparatus in the kidney releases renin, an enzyme which cleaves angiotensinogen into angiotensin I (Ang I), an inactive decapeptide. Subsequently, Ang I will be turned into angiotensin II (Ang II) by the angiotensin I-converting enzyme (ACE) [7], mainly expressed at the surface of endothelial cells. Ang II, an octapeptide, is the main endogenous ligand of the RAS, and binds to two main receptors: the angiotensin II type 1 receptor (AT_1_) and the angiotensin II type 2 receptor (AT_2_), both belonging to the G protein-coupled receptor (GPCR) family [8,9]. The hydrolysis of Ang II [10] by the angiotensin II-converting enzyme 2 (ACE2) produces the heptapeptide Ang-(1-7). Ang-(1-7) mediates signaling via the Mas receptor (MasR) and the Mas-related G protein-coupled receptor member D (MrgD receptor) (Figure 1). In addition, decarboxylation of Ang-(1-7) transforms it into alamandin, which is also able to bind to the MrGD receptor. These two axes have been the subject of many recent reviews, and will not be discussed in this article [11,12,13]. The AT_4_ receptor, whose ligand is Ang IV, has been identified as a transmembrane enzyme, insulin-regulated membrane aminopeptidase (IRAP) [14]. In addition to its vasorelaxant effect in cerebral [15] and renal [16] vascular beds, the AT_4_ receptor seems to be involved in memory and in Alzheimer’s disease [17,18].

The two main receptors for Ang II, the AT_1_ and AT_2_ receptors, share a similar affinity for Ang II [9] but exert opposite actions [19,20]. These characteristics suggest that the stimulation of RAS and Ang II production may lead to physiological responses that directly reflect the functional balance between AT_1_ and AT_2_ receptors. From a systemic point of view, most of the known effects of RAS activation (elevated blood pressure, water and sodium retention, aldosterone release…) are subsequent to AT_1_ receptor activation, as the expression and activity of AT_2_ receptor seem too low to counteract AT_1_ receptor stimulation. Apart from the systemic RAS, many studies have shown the existence of a localized expression of RAS components in various tissues. For instance, Campbell and Habener in 1986 measured angiotensinogen mRNA levels in 17 different organs in rats; brain, spinal cord, aorta, and mesentery levels were similar to hepatic levels, whereas the levels were lower in the kidney, adrenal, atria, lung, large intestine, spleen, and stomach [21]. Moreover, in humans, the expression of angiotensinogen mRNA is also found in different organs as indicated by the Human Protein Atlas.

These results indicate that many of the tissue-specific actions of angiotensin II may be mediated by local tissue RAS, independently of the circulating RAS. Thus, at this tissue level, a possible balance of these AT_1_/AT_2_ receptor appears to be of major importance in the regulation of RAS activity, and the fine tuning of this AT_1_/AT_2_ balance seems critical.

The objective of this review is to emphasize the importance of this local functional balance between AT_1_ and AT_2_ receptors in physiological and pathophysiological processes. After a brief analysis of the structure and signalization of these two receptors, which have been recently reviewed [22,23], we will highlight the importance of the AT_1_/AT_2_ equilibrium. We will not illustrate this point through a systematic review, but through several examples chosen to emphasize the ubiquitous aspect of this major regulation of physiological functions. As we are interested in the vascular and cerebrovascular effects of this system, these will be described first, followed by the cell cycle/cancerization process and wound healing. In the last chapters, we will discuss the mechanisms of this AT_1_/AT_2_ balance, as well as different perspectives to be considered in order to modulate it.

## 2. AT_1_ and AT_2_ Angiotensin II Receptors

### 2.1. AT_1_ Receptor

The AT_1_ receptor is responsible for vasoconstriction, cell growth and proliferation, oxidative stress, inflammation, and also hypertrophy and hyperplasia [24,25]. Most of these actions result from the activation of intracellular signaling pathways involving several phospholipases and kinases (see below).

This receptor is expressed in several organs such as artery walls (smooth muscle cells), wherein it is highly expressed [26], but also in the heart (cardiomyocytes) [27], kidney (glomeruli, proximal convoluted tubules) [28] and brain (neurons, microglia cells) [29]. Two isoforms, the AT_1A_ and AT_1B_ receptors [30], have been identified in rodents, showing a sequence homology of more than 96%, and identical functions. In humans, only one isoform has been identified.

The AT_1_ receptor is a GPCR classically described to activate the phospholipase C (PLC) via G_q_ protein, although it also interacts with G_i_, G_12/13_, and G_s_ proteins [20]. 

#### 2.1.1. Structure

Advances in protein crystallization have led to the elucidation of crystalline structures for GPCRs, providing insight into activation and signaling mechanisms. As GPCRs are often involved in diseases, these crystalline structures also pave the way for structured drug design. The emergence of X-ray crystallography allowed the first crystalline structure of the AT_1_ receptor to be co-crystallized with an angiotensin II receptor blocker (ARB) [31].

AT_1_ receptor is a member of the seven transmembrane or GPCR family. Its sequence of 359 amino acids includes three *N*-glycosylation sites (that enable the proper folding of the receptor and account for its trafficking to the membrane) and four cysteine residues at the extracellular regions [20] (Figure 2). In addition to the two cysteines involved in a disulfide bridge between the first and second extracellular loops as for all GPCRs, the AT_1_ receptor contains an additional pair of extracellular cysteine residues. These cysteine residues are located on the N-terminal part and the third extracellular loop, and thus form a second disulfide bridge responsible for maintaining the conformation of the AT_1_ receptor and its binding to Ang II [32]. The cytoplasmic region of the receptor, composed of three intracellular loops and the C-terminal tail, contains sites that can be phosphorylated by several serine/threonine kinases, such as protein kinase C (PKC), and also has four cysteine residues which are involved in disulfide bridges [20].

The AT_1_ receptor can adopt three different conformations that directly influence its activity [33]. The inactive conformation is stabilized by ARBs and does not lead to any downstream signaling. The “canonical active” conformation is observed after the binding of the endogenous ligand (Ang II), and allows the activation of many different signaling pathways (see below). The binding of Ang II results in a movement of the seventh transmembrane domain on the intracellular side, allowing the recruitment of G proteins or of β-arrestin. Finally, the “active alternative” conformation (incomplete movement of the seventh transmembrane domain on the intracellular side) prevents G protein coupling to the DRY motif of the receptor (Figure 1), and only allows recruitment and stimulation of the β-arrestin signaling pathway [33,34].

**Figure 2 molecules-28-05481-f002:**
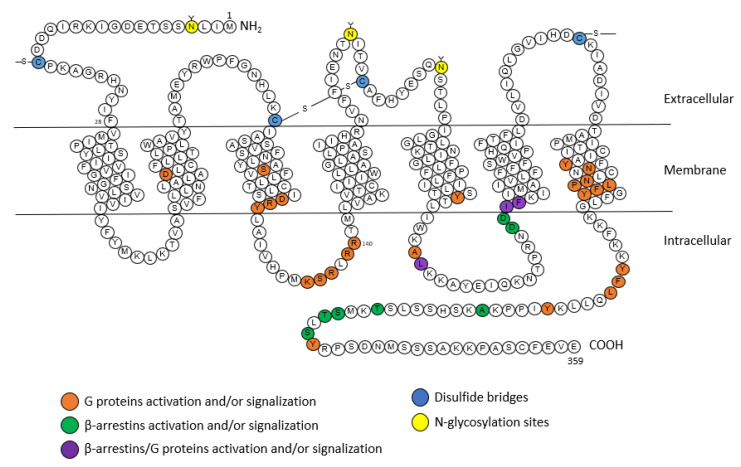
Snake plot of the rat AT_1_A receptor (modified from [20,35,36]). Orange: amino acids for the activation of G proteins, green: amino acids for the activation of β-arrestins, purple: amino acids for the activation of G proteins and β-arrestins, blue: disulfide bridges, yellow: N-glycosylation sites. A: alanine; C: cysteine; D: aspartic acid; E: glutamic acid; F: phenylalanine; G: glycine; H: histidine; I: isoleucine; K: lysine; L: leucine; M: methionine; N: asparagine; P: proline; Q: glutamine; R: arginine; S: serine; T: threonine; V: valine; W: tryptophane; Y: tyrosine.

#### 2.1.2. Signaling

##### G Protein Pathway

As evoked above, the AT_1_ receptor, once activated by Ang II, adopts the canonical active conformation, allowing its coupling to G proteins (Figure 3). 

The activation of phospholipase C (PLC) is subsequent to Gα_q_ protein activation. This activation releases the α_q_ and β_γ_ subunits of the Gα_q_ protein. The α_q_ subunit activates the PLC, leading to the hydrolysis of phosphatidyl-inositol-4,5-diphosphate (PIP_2_) into inositol-1,4,5-trisphosphate (IP_3_) and diacylglycerol (DAG) [37]. The production of IP_3_ induces a release of calcium (Ca^2+^) from the endoplasmic reticulum, and then Ca^2+^ complexes with calmodulin; this then activates myosin light-chain kinase (MLCK). Phosphorylation of myosin light chains (MLC) leads to muscle cell contraction and thus vasoconstriction. Following the elevation in Ca^2+^ induced by the PLC, the phospholipase D (PLD) is activated, causing hydrolysis of phospholipids (mainly phosphatidylcholine) to generate phosphatidic acid, which is itself transformed into DAG [38]. Subsequently, DAG activates PKC, which induces the activation of PLC; thus, this positive feedback allows the continuous maintenance of PLC activity [39]. The Gα_q_ subunit can also stimulate cell growth and proliferation through the activation via phosphorylation of several downstream proteins such as mitogen-activated protein kinases (MAPKs), janus kinases (JAKs), and transcription signal transducer and activator (STAT) proteins [40]. 

AT_1_ receptor stimulation also induces the activation of phospholipase A_2_ (PLA_2_) via the activation of the Gα_q_ protein. Once activated, PLA_2_ allows the release of arachidonic acid from membrane phospholipids [41]. Arachidonic acid is then transformed into eicosanoids such as prostaglandins or thromboxane [42] by cyclooxygenases or lipooxygenases. Several of these eicosanoids play a role in Ang II-induced contraction, whereas others (PGI2, PGE2) oppose it [42]. The AT_1_ receptor may also recruit other G proteins, such as the G_12/13_ protein, involved in the activation of the RhoA/ROCK (Rho-associated protein kinase) signaling. These ROCKs are serine-threonine kinases with targets involved in the regulation of contractility [43,44].

##### β-Arrestins

After stimulation by Ang II, the AT_1_ receptor is phosphorylated on its intracellular C-terminal serine and threonine residues by GPCR kinases (GRKs). This phosphorylation increases the affinity of β-arrestin-1 and β-arrestin-2 equally for the AT_1_ receptor. The β-arrestins’ recruitment is known to inhibit G protein-induced signaling by interfering with the conformation of the receptor; it is also known to initiate, with the help of clathrins and AP-2 adaptor proteins, the internalization and sequestration of the AT_1_ receptor coupled to its ligand, as well as the membrane recycling of the receptors [34,45].

Originally, arrestins were identified as central players in the desensitization and internalization of GPCRs. In addition to modulating GPCR signaling, in 1999, Robert Lefkowitz’s team showed that β-arrestins can also initiate a second wave of signaling [46]. Other studies, such as that of Tohgo, have shown that overexpression of β-arrestin-1 or β-arrestin-2 leads to a decrease in inositol-phosphate (IP) production following AT_1_ receptor stimulation [47]. In addition to the modulation of GPCR signaling through desensitization and internalization, β-arrestins also act as signaling scaffolds for various signaling pathways.

For example, β-arrestins are able to recruit at the plasma membrane proteins belonging to the Src family (internalization of the receptor is not necessary) [48]. These Src allow the activation of different kinases such as ERKs, leading to a decrease in the production of IP following the stimulation of AT_1_ receptor [47]. 

##### NADPH

Griendling et al. were the first to demonstrate the implication of nicotinamide adenine dinucleotide phosphate (NADPH) in oxidative stress mediated by the AT_1_ receptor [49]. Using rat aortic VSMCs, they showed that treatment of VSMCs with Ang II for 4–6 h caused a nearly threefold increase in intracellular O_2_^-^ consumption. Ang II stimulates the activity of NAD(P)H oxidase (NOX), and thus generates reactive oxygen species (ROS) [49]. 

NOX comprises five subunits, and in the absence of stimulation, some of its subunits are cytosolic, while others are membrane-bound [50]. Ang II, via processes involving several players such as c-Src, PLD, PKC, PI3K, and transactivation of EGFR (epidermal growth factor receptor), induces phosphorylation of the p47phox subunit, which causes the formation of a complex between cytosolic subunits, followed by transfer to the membrane, wherein the complex associates with membrane subunits to give the active form of the oxidase [50]. This will lead to the production of reactive oxygen species (ROS) such as H_2_O_2_ or superoxide. These ROS are able to activate transcription factors such as activator protein-1 (AP-1) and nuclear factor kβ (NF-kβ), which will induce the expression of pro-inflammatory genes [44].

**Figure 3 molecules-28-05481-f003:**
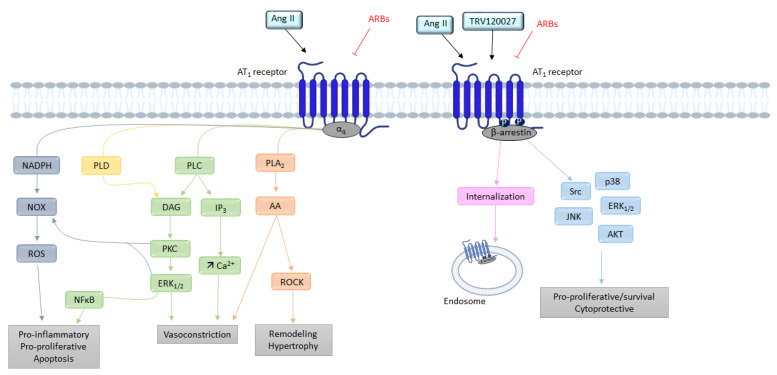
Overview of the different signaling pathways following AT_1_ receptor activation. AA: arachidonic acid; Ang II: angiotensin II; ARBs: angiotensin receptor blockers; AT_1_: angiotensin II type 1 receptor; AT_2_: angiotensin II type 2 receptor; Ca^2+^: calcium; DAG: diacylglycerol; ERK_1/2_: extracellular signal regulated kinase ½; IP_3_: inositol triphosphate; JNK: c-Jun N-terminal kinase; NFκB: nuclear factor kappa B; NOX: NADPH oxidase; PLA2: phospholipase A2; PLC: phospholipase C; PLD: phospholipase D; PKC: protein kinase; ROCK: Rho-associated protein kinase; ROS: reactive oxygen species; TRV120027: β-arrestin-biased AT_1_ agonists.

### 2.2. AT_2_ Receptor 

The AT_2_ receptor was initially not widely studied because of its low abundance in tissues making its study more difficult. However, it seems to play an important role in the development of the circulatory system, in particular by allowing the differentiation of precursor cells into smooth muscle cells during gestation, thus influencing the structure and function of blood vessels [19]. After birth, AT_2_ receptors are restricted to certain tissues such as those of the brain, heart, vascular endothelium, kidney, uterus, and ovary [20]. Besides their decrease in locomotion and exploratory behavior associated with a decrease in spontaneous movements and rearing activity, AT_2_ receptor-KO mice suffer from impaired drinking response to water deprivation [51,52]. Moreover, AT_2_ receptor expression is upregulated during inflammation, and its stimulation reduces organ damage [53].

#### 2.2.1. Structure

The first crystalline structures of the AT_2_ receptor were published in 2017 [54] demonstrating commonalities such as an extracellular loop 2 (ECL2) β-hairpin conformation. 

The AT_2_ receptor is composed of 363 amino acids and shares 34% homology with the AT_1_ receptor. The AT_2_ receptor has seven transmembrane domains, an extracellular amino terminus and an intracellular carboxy terminus (Figure 4) [9]. However the AT_2_ receptor seems to have unique structural and functional differences, unlike other GPCRs (including AT_1_ receptor), the AT_2_ receptor is not internalized after stimulation by its endogenous agonist (Ang II) and this stimulation does not lead to the binding of stable GTP analogues [20].

The AT_2_ receptor also has a nanomolar affinity for Ang II, similar to that of the AT_1_ receptor (Table 1). At the N-terminal part AT_2_ receptor has five n-glycosylation sites and 14 cysteine residues [20]. The second intracellular loop consists of a potential PKC phosphorylation site. The cytoplasmic tail contains three consensus PKC phosphorylation sites but also a phosphorylation site for the cyclic AMP-dependent protein kinase [55]. The third intracellular loop of the receptor is involved in coupling to G_i_ proteins [56], and thus is responsible for inhibiting AT_1_ receptor-dependent IP3 production. This third loop is also essential for the signal transduction of AT_2_ receptor via MAPK, and extracellular signal-regulated kinase (ERK) inactivation [55].

**Figure 4 molecules-28-05481-f004:**
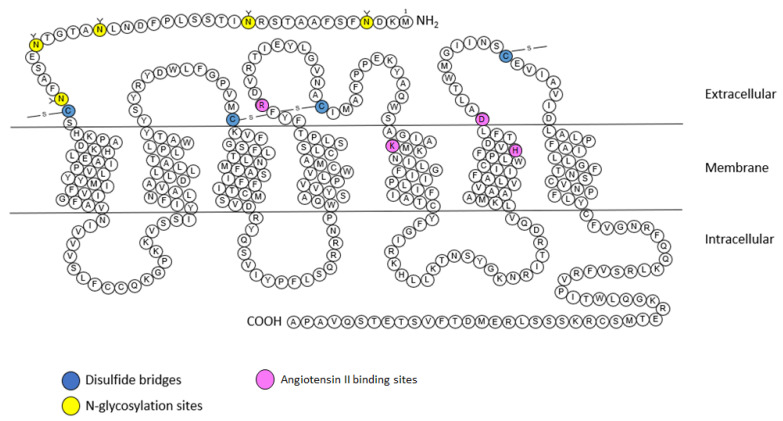
Snake plot of the rat AT_2_ receptor (modified from [57,58,59,60]). Blue: disulfide bridges; yellow: *N*-glycosylation sites. Purple: Angiotensin II binding sites. A: alanine; C: cysteine; D: aspartic acid; E: glutamic acid; F: phenylalanine; G: glycine; H: histidine; I: isoleucine; K: lysine; L: leucine; M: methionine; N: asparagine; P: proline; Q: glutamine; R: arginine; S: serine; T: threonine; V: valine; W: tryptophane; Y: tyrosine.

#### 2.2.2. Signaling

The effects produced by stimulation of the AT_2_ receptor are the result of the activation of intracellular signaling pathways different from those of the AT_1_ receptor [61]. It is interesting to note that unlike most other GPCRs, the AT_2_ receptor does not associate with β-arrestins [62]. Although the AT_2_ receptor-induced signaling pathways are not yet well understood, they involve in particular the NO, the bradykinin (BK), and the activation of several proteins with tyrosine phosphatase activity [63] (Figure 5).

##### G Protein Pathway

Before being cloned and identified, this receptor was considered independent of any interaction with G proteins [64]. Nevertheless, several biochemical and functional studies have indicated that the AT_2_ receptor may recruit G_i_ [19,56], thereby resulting in activation of the NO-cyclic GMP (cGMP)-protein kinase G_i_ pathway [65]. Subsequently cGMP activates PKG_i_, which dephosphorylates myosin light chains via MLCK, thus preventing calcium from leaving the endoplasmic reticulum. 

Moreover, G_i_ recruitment leads to downstream activation of various phosphatases, such as MAPks, SH2-domain-containing phosphatase 1 (SHP-1), and serine/threonine phosphatase 2A (PP2A), resulting in the opening of delayed rectifier K^+^ channels and inhibition of T-type Ca^2+^ channels [40]. The activation of G_βγ_ subunits by the AT_2_ receptor can induce the release of arachidonic acid via the PLA_2_. The metabolites produced by arachidonic acid (prostaglandins or thromboxane) appear to contribute to AT_2_ receptor-mediated vasodilation [66].

We have seen previously that AT_1_ receptor-mediated vasoconstriction is mediated by the RhoA/ROCK pathway, among others,. Studies by Savoia have shown that the AT_2_ receptor decreases the activation of the RhoA/ROCK pathway [67], and that this decrease seems to be associated with an increase in the expression of PKGI, which inactivates RhoA by phosphorylating it [68].

##### Bradykinin

Siragy et al. has suggested that AT_2_ receptor activity is mediated by stimulation of bradykinin (BK) production [69]. This hypothesis was later confirmed by showing that AT_2_ receptor inhibits the activity of Na^+^/H^+^ exchangers, resulting in acidification of the cell environment, which ultimately results in the release of BK [70]. AT_2_ receptor-dependent stimulation of BK receptors (B_2_ receptor) seems to activate protein kinase A (PKA), which phosphorylates eNOS [71]. Furthermore, the proximity of the two receptors allows heterodimerization of AT_2_ receptors and B_2_ receptors, increasing the production of cGMP and NO [72]. In addition to these pathways, the AT_2_ receptor is capable of inducing NO-independent vasodilation. Indeed, AT_2_ receptor is able to induce hyperpolarization of smooth muscle cells by inducing vasodilation mediated by potassium channels [73]. Inhibition of these channels would abolish this vasodilation.

In conclusion, via activation of the AT_2_ receptor, Ang II leads to vasodilation, anti-proliferative, and pro-apoptotic effects, meaning the effects of AT_1_ receptor activation can be counteracted [74].

**Figure 5 molecules-28-05481-f005:**
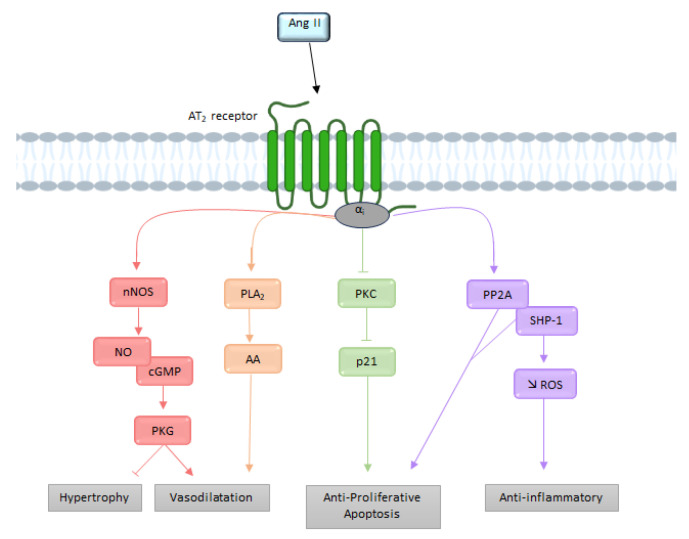
Overview of the different signaling pathways following AT_2_ receptor activation. AA: arachidonic acid; Ang II: angiotensin II; AT_2_: angiotensin II type 2 receptor; cGMP: cyclic guanosine monophosphate; NO: nitric oxide; nNOS: neuronal nitric oxide synthase; PLA2: phospholipase A2; PKC: protein kinase; PP2A: serine/threonine phosphatase 2A; ROS: reactive oxygen species; SHP-1: SH2-domain-containing phosphatase 1.

## 3. The Functional AT_1_/AT_2_ Receptors Balance

We have seen previously that the RAS is expressed in several tissues. We will now discuss, through four different examples, the physiological and pathophysiological implication of the AT_1_/AT_2_ functional balance [75]. Several phenomena may interfere with this balance, such as a change in receptors’ expression, or an increase of the production of Ang II.

### 3.1. Systemic Cardiovascular Impact

Most of the known physiological and pathological actions of Ang II are mediated by AT_1_ receptors. One of the first systemic effects of the AT_1_ receptor to be discovered was the regulation of blood pressure. A decrease in blood pressure will induce a decrease in renal perfusion, leading to a release of renin. The renin will then cleave angiotensinogen (in excess) into Ang I, which will in turn be cleaved into Ang II, which will cause vasoconstriction of the vessels, thereby increasing peripheral resistance with the effect of increasing blood pressure [76]. In addition, Ang II can also increase blood pressure via a decrease in renal excretion of water and sodium [6]. 

In 1999, Horiuchi and his team showed that AT_2_ receptors have opposite effects to those of AT_1_ receptors [19]. AT_2_ receptors also play an important role in the regulation of renal function, particularly with respect to Na^+^ and water excretion, leading to a reduction in blood pressure. However, studies have shown that stimulation of the AT_2_ receptors by an agonist does not always lead to a reduction in blood pressure, but that it depends on the model, as shown in this review [77].

In physiological conditions, the AT_1_ receptor’s effects predominate during Ang II stimulation, due to its higher abundance in tissues.

### 3.2. AT_1_/AT_2_ Balance in the Brain and Cerebral Circulation

#### 3.2.1. Cerebral Circulation

The discovery of the presence of cerebral Ang II in 1971 revealed a cerebral RAS in dogs and rats [78]. Subsequently, AT_1_ and AT_2_ receptors were localized in cerebral blood vessels in rats [51], and then in the human brain [79].

In a rat cranial window model, Vincent et al. showed that Ang II-induced vasoconstriction of cerebral arterioles was abolished when an AT_1_ receptor antagonist was used, resulting in vasodilation, which was itself abolished when AT_2_ receptor antagonists were used [80]. In physiological conditions, the AT_1_ receptor will allow vasoconstriction of cerebral arterioles while the AT_2_ receptor has the opposite effect. Moreover, in the same cranial window model, our team showed that the AT_1_ receptor was involved in the structural remodeling of cerebral arterioles in hypertensive rats [81]. This remodeling of cerebral arterioles induces a decrease in the internal diameter of vessels that is reversed when AT_1_ receptor antagonists are used [82].

The AT_1_ and AT_2_ receptors and changes in the AT_1_/AT_2_ equilibrium are particularly important contributors to the regulation of cerebral circulation. Indeed, the vasoconstriction of cerebral arterioles observed during Ang II stimulation is the sum of AT_1_ receptor-dependent vasoconstriction and AT_2_ receptor-dependent vasodilation in physiological conditions.

Importantly, under pathological conditions with deregulation of the cerebral circulation, such as cerebral ischemia, altered function and expression of RAS components in the cerebral vasculature is observed. During ischemic stroke, increased AT_1_ receptor-dependent vasoconstriction of cerebral vessels has been shown in the presence of Ang II, despite decreased AT_1_ receptor gene expression [83]. To counteract this, it has been shown that AT_2_ receptor gene expression is increased following stroke, but AT_2_ receptor protein expression remains unchanged in the middle cerebral arteries, which limits the beneficial impact that AT_2_ receptor agonists may have [84]. 

#### 3.2.2. Cardiovascular Regulation

The two receptors are expressed inside or near the medulla oblongata, the brain region which regulates cardiac rhythm and blood pressure. They are exclusively found in the neurons rather than the glia. 

The localization of AT_1_ receptors in the brain has been determined primarily by receptor autoradiography [32,85,86]. These studies demonstrated a wide distribution of AT_1_ receptors’ expression in the brain, including several regions involved in cardiovascular regulation. This distribution of AT_1_ receptors in the brain was confirmed by the development of transgenic mice expressing the AT_1_ receptor fused with eGFP [87]. High densities of AT_1_ receptors are found in the subfornical organ (SFO), the paraventricular nucleus (PVN), the area postrema (AP), the nucleus of the solitary tract (NTS), and the rostral ventrolateral medulla (RVLM) [87]. AT_1_ receptors are almost exclusively localized in neurons rather than in microglia, or astrocytes in these cardiovascular control centers [88]. Furthermore, the AT_1_ receptor appears to be predominantly expressed on glutamatergic neurons [89]. 

Similarly, the results obtained with AT_2_ receptor-eGFP mice demonstrate the localization of AT_2_ receptor-positive cells in or near brain areas that directly influence sympathetic output and blood pressure control. For example, the use of this mouse not only confirmed the presence of AT_2_ receptor-containing neurons in the intermediate NTS, but also demonstrated that these neurons are predominantly GABAergic [90]. The AT_2_ receptor–eGFP mouse also revealed a high concentration of AT_2_ receptor-positive neurons in the AP, as well as AT_2_ receptor neuronal fibers in the RVLM and PVN [90]. Furthermore, a number of AT_2_ receptor-positive neuronal fibers and terminals in the PVN were found to be derived from the AT_2_ receptor-containing GABAergic neurons that surround this nucleus [89]. 

It has been shown that AT_1_ receptors can directly influence the neurons of the cardiovascular centers to induce sympatho-excitation and increase blood pressure, indeed; the mRNA expression levels of the AT_1_ receptor in the RVLM and the NTS were higher in hypertensive rats in comparison to normotensive rats [91]. Furthermore, the administration of the AT_1_ receptor antagonist losartan into the brain reduces blood pressure in hypertensive rats [91]. The first evidence that stimulation of the brain of AT_2_ receptors lowers blood pressure was demonstrated by the fact that AT_1_ receptor-mediated pressor responses to Ang II were amplified in the presence of the AT_2_ receptor antagonist PD123319 [92]. Furthermore, the stimulation of RVLM AT_2_ receptor by a specific agonist (CGP42112A) results in a drop in blood pressure [93].

#### 3.2.3. Neuroinflammation

Inflammation is another example of the dysregulation of the AT_1_/AT_2_ receptors’ balance in the brain. Cells present in the brain such as astrocytes or microglial cells are the main sources of inflammation. In general, the activation of the AT_1_ receptor in macrophages leads to the activation of the pro-inflammatory axis of the RAS, while activation of the AT_2_ receptor promotes the activation of an anti-inflammatory axis [94,95]. 

In contrast, in pathological conditions such as inflammation, these receptors (and in particular, AT_1_/NOX signaling) are upregulated. NOX-derived superoxides are amplified by the activation of NF-kβ and the RhoA/ROCK pathway, leading to the production of ROS. In addition, through a feedback mechanism, activation of the RhoA/ROCK pathway enables increased expression of the AT_1_ receptor via NF-kβ [96]. To compensate for this mechanism, it was shown by Rompe et al. that the AT_2_ receptor can induce the dephosphorylation of I-κB (NF-κβ inhibitor), thus allowing it to bind to NF-κβ to prevent its nuclear translocation (and transcription of the AT_1_ receptor) [53].

However, it has been shown that although AT_2_ receptor expression is generally upregulated, as the AT_1_ receptor as a compensatory mechanism, this phenomenon attenuates during aging. For example, in young and healthy brains of rats, and as previously described, there is a balance between the pro-oxidative and pro-inflammatory axis induced by AT_1_ receptor, and between the protective axis mediated by the AT_2_ receptor [97]. Aging causes an overactivation of the pro-inflammatory axis of the RAS, while the protective axis is unchanged. Indeed, a significant increase in NOX activity and levels of the pro-inflammatory cytokines was observed in aged rats, which revealed a pro-oxidative and pro-inflammatory state in the aged substantia nigra [97]. Moreover, aged rats also showed upregulation of AT_1_ receptor expression, which was inhibited by administration of candesartan (an AT_1_ receptor blocker), and downregulation of AT_2_ receptor expression [98].

Through these examples, we have seen different mechanisms of deregulation of the AT_1_/AT_2_ balance in the brain. Although the upregulation of the two receptors can partly explain this deregulation, it is certainly not the only element involved in the appearance of pathologies, as we will see in the following example.

### 3.3. Cellular Cycle

The proliferation of normal cells is regulated by kinases called cyclin-dependent kinases (CDKs). The main actors in this cell cycle are cyclins, which regulate CDKs, enabling cells to progress through the cell cycle [99]. We have seen previously that AT_1_ receptors promote cell proliferation pathways, in particular via the MAPK pathway, which allows the expression of cyclin and CDKs, itself allowing the advancement of cells in the cell cycle (Figure 6) [100]. Diep et al. studied the expression of cell cycle proteins in Ang II-infused rats. A significant increase in ^3^H-thymidine incorporation in the mesenteric arteries was observed, reflecting the entry of cells into DNA replication phase (S phase). Furthermore, this incorporation was associated with high expression of cyclin D_1_ and CDK4 [101]. The use of losartan in these animals completely abolished ^3^H-thymidine incorporation and restored the expression level of cyclin D_1_ and CDK4.

Conversely, AT_2_ receptors are involved in the activation of anti-proliferative pathways [102]. In human aortic endothelial cells and after stimulation with C21 (an AT_2_ receptor agonist), an activation of p53 protein was observed [103]. This protein is a key tumor suppressor which can lead either to a transient arrest of cell proliferation (to repair DNA damages, for example) or to an irreversible arrest of cell proliferation called senescence, leading to cell death [104]. When p53 is not activated, histone deacetylase-1 (HDAC1) deacetylates p53, resulting in its degradation; in contrast, HDAC1 inhibitors (such as the vorinostat) sensitize cells to apoptosis by increasing p53 acetylation. This AT_2_ receptor-coupled HDAC1/p53 signaling pathway appears to have a role in physiological apoptosis and cell turnover involving p53 [103].

The AT_1_ receptor directly induces cell hypertrophy, notably through the MAPk pathway, but also through the β-catenin pathway [105,106]. Cyclin D_1_ will enable the transition from the G_0_ to the G_1_ phase of the cell cycle. The next expected step for cells in G_1,_ is the S phase, leading to cell proliferation. However, Geisterfer et al. reported that Ang II induces hypertrophy, but not proliferation, in confluently cultured rat aortic smooth muscle cells [107]. Subsequently, several studies revealed that Ang II stimulates the expression of immediate early genes exclusively expressed in the G_1_ phase (c-myc and PDGF (platelet-derived growth factor)) [108]. Thus, these genes, once expressed after Ang II stimulation, will allow the return of resting cells to the G_1_ phase, and not necessarily progression to S phase, which consists of G_1_ phase arrest and hypertrophy [109].

In esophageal adenocarcinomal cells (EACs), Fujihara et al. showed that telmisartan induces antitumoral effects in EAC, both in vitro and in vivo. Following inhibition of the AT_1_ receptor by telmisartan, cell cycle arrest in the G_0_/G_1_ phase is induced via the Akt/mTOR pathway in EAC cells [110]. Similar results were obtained in breast cancer and cholangiocarcinoma cells [111]. Moreover, this stop in the cell cycle was accompanied by a high decrease in cell cycle-related proteins, such as cyclin E, cyclin D_1_, and their catalytic sub-units, Cdk4 and Cdk6. These experiments show that increased cell proliferation in cancer is at least partially related to actions mediated by the AT_1_ receptor. In contrast, overexpression of AT_2_ receptors in SMMC7721 cells was shown to reduce S-phase cells and increase G_1_-phase cells via suppression of CDK4 and cyclin D_1_ expression, thereby reducing proliferation [112].

Many studies have reported that an increase in the presence of AT_1_ and AT_2_ receptors is found in different types of cancer, and is directly linked to a worse prognosis in terms of tumor aggressiveness [113,114,115]. The overexpression of the AT_1_ receptor has been demonstrated in several in vitro models, such as mammary carcinoma cells in culture, pancreatic adenocarcinoma cells, and hepatocarcinoma cells, but also in vivo in various tumors including estrogen receptor positive breast cancers, glioblastoma, and ovarian cancers [116,117]. Despite its anti-proliferative effect, the AT_2_ receptor has also been shown to be overexpressed by several cancers such as astrocytomas [113] and lung tumors [118] in vivo. In an astrocytoma model, it was shown that out of 133 tumors, 10% of low-grade tumors were positive for the AT_1_ receptor, versus 67% for high-grade tumors; 17% of low-grade tumors were positive for the AT_2_ receptor, versus 53% for high-grade tumors [113].

It has been shown in several models of cancer cells overexpressing the AT_2_ receptor that this overexpression promotes apoptosis of these cells [112,119,120]. In addition, the effects of C21 were studied in prostate cancer (using human LNCaP prostate cancer cells and prostate adenocarcinoma transgenic rats) and leiomyosarcoma cells. In both studies, the authors were able to find antiproliferative and pro-apoptotic effects of the AT_2_ receptor [121,122].

### 3.4. Wound Healing

Several studies reported dynamic changes in angiotensin receptor expression during the different phases of wound healing [123,124].

The expression of AT_1_ and AT_2_ receptors in the skin of young rats was first shown in 1992 by Viswanathan and Saavedra [125]. AT_1_ and AT_2_ receptors are expressed in human fibroblasts, keratinocytes, and vascular endothelial cells. Both AT_1_ and AT_2_ receptors are found in myofibroblasts and keratinocytes in rodents [126]. RAS components are present in the epidermal and dermal layers, but also in subcutaneous fat tissues, in microvessels, and in appendages such as hair follicles [126,127]. However, expression of RAS components in skin has also been demonstrated at the protein level, with results confirmed at the mRNA level [128].

Regarding the functional role of the RAS in skin physiology, a recent study by Jiang et al. reported that Ang II promotes differentiation of keratinocytes from bone marrow-derived mesenchymal stem cells (BMdSC) under physiological conditions [129]. Moreover, Ang II has been shown to increase vascular permeability to recruit inflammatory cells and to induce angiogenesis [130]; the AT_1_ receptor promotes migration, while the AT_2_ receptor inhibits it.

During wound healing, an organism will regulate the expression levels of AT_1_ and AT_2_ receptors, enabling a response to Ang II that is adapted to the situation. Immediately after wounding, an increase in both AT_1_ and AT_2_ receptor expression is observed, which seems slightly delayed and weaker for AT_2_ receptors [131]. In cultured keratinocytes, this regulation is detectable at the mRNA level 1 h after wounding, but the protein expression of AT_1_ receptor peaks at 3 h, and that of AT_2_ receptor peaks at 12 h after wounding [126]. This specific early increase in AT_1_ receptors could play a role in promoting blood clotting, initiating the inflammatory phase and inducing re-epithelialization by stimulating keratinocyte proliferation and migration [132,133].

In vivo, the wound healing process leads to an increase in receptor expression; this is higher for the AT_1_ receptor than for the AT_2_ receptor during the early phases of wound closure. Subsequently, there is a decrease in the expression of both receptors during the inflammation process, followed by an increase during re-epithelization.

Finally, during the last phase (remodeling), an increase in AT_1_ and AT_2_ receptors has been demonstrated, but this time with a dominance of the AT_2_ receptor over the AT_1_ receptor (Figure 7) [123,133].

This is consistent with what is known about both receptors, since in the early phases of wound healing and re-epithelization, a pro-proliferative action of AT_1_ receptors is required to allow wound closure. Therefore, it has been shown that in mice KO for AT_1_ receptors and rats treated with an AT_1_ receptor antagonist, wound closure was delayed [134,135]. In contrast, in AT_2_ receptor KO mice, re-epithelization was accelerated. These results support that during the wound healing process, the antiproliferative effect of AT_2_ receptors is complementary to the pro-proliferative effects of AT_1_ receptors under physiological conditions. 

During the remodeling phase of wound healing, AT_2_ receptor expression is stronger than AT_1_ receptor expression. The antifibrotic properties seem essential for the formation of a resistant scar tissue. Indeed, in AT_2_ receptor-KO mice, the skin breaks under lower tension than in wild-type mice [135]. This increased expression of the two receptors contributes to the localized Ang II action in the wounded area rather than in the unaffected skin. Thus, the changes in the AT_1_/AT_2_ receptor ratio observed in tissue repair may cause a switch in the dominating subtype, and consequently a change in the response to Ang II [136]. 

The formation of hypertrophic scars or keloids is a recurrent problem resulting from an insufficient control of proliferative and fibrotic processes in wound healing [137]. Indeed, it has been shown that it is the overactivated cutaneous RAS that is involved in this process via the AT_1_ receptor [20,138]. Indeed, these receptors, once stimulated by Ang II, are known to act in a pro-fibrotic manner. Studies have also shown that the level of Ang II and AT_1_ receptor expression were increased in hypertrophic scars and keloids in both rodents and humans. As a result, several experiments were conducted to inhibit the profibrotic effect of AT_1_ receptor by using ARBs to prevent or treat hypertrophic scars and keloids in preclinical models. All these studies, regardless of the species, resulted in a reduction in scar size [139,140].

As evoked above, AT_1_ and AT_2_ receptors have the same affinity for Ang II, but induce opposite physiological responses. Thus, the physiological response to Ang II production will reflect the functional balance between these two receptors. In the next section, we will discuss the different mechanisms that may contribute to the fine tuning of this functional AT_1_/AT_2_ receptor balance.

## 4. Mechanisms Regulating the AT_1_/AT_2_ Functional Balance 

### 4.1. Functional Opposition vs. Expression Level

The response to Ang II results from the balance between the effects of each of these receptors. As we have seen previously, the effects of the AT_1_ receptor predominate because it is the most abundant.

This functional balance was confirmed in vivo using knockout mice for either the AT_1_ receptor or AT_2_ receptor. In AT_1A_ receptor KO mice, there was an absence of hypertensive response following Ang II injection, which is normally observed in wild-type mice. In addition, systemic blood pressure was markedly decreased in these mice [141]. In the same AT_1A_ receptor KO mice and after treatment with an AT_1_ receptor antagonist, a decrease in blood pressure was observed in mice pretreated with ACE, thus showing that AT_1B_ receptors also seem to play a role in the regulation of blood pressure. In contrast, in AT_2_ receptor KO mice, vasoconstriction to Ang II is more important than in wild-type mice [142]. Thus, in cerebral arteries, the effects observed are the sum of constrictor effects mediated by the AT_1_ receptor and dilator effects mediated by the AT_2_ receptor.

Furthermore, as we have seen in Section 3.4, during the wound healing phenomenon, there is a modification of the expression of the receptors according to the phase of the process: a stronger expression of the AT_1_ receptor at the time of re-epithelization, and a stronger expression of the AT_2_ receptor at the time of the remodeling phase.

The expression level of the receptors is also related to the mechanisms that regulate them. At the AT_1_ receptor level, overexpression of ATRAP (AT_1_ receptor-associated protein) inhibits AT_1_ receptor-dependent PLC activation [143], inositol phosphate production, and cell proliferation [144], indicating that ATRAP acts as a negative regulator of AT_1_ receptor signaling [145]. In addition, overexpression of ATRAP decreases membrane expression of the AT_1_ receptor due to its increased internalization [146]. In contrast, expression of ARAP-1 (ankyrin repeat and pleckstrin homology domain-containing protein 1) restores receptor membrane expression [147]. Moreover, at the renal level, overexpression of ARAP1 in mice causes hypertension and renal hypertrophy, effects that are suppressed by an ARB, suggesting that ARAP1 potentiates AT_1_ receptor signaling [148].

The AT_2_ receptor interacts with ATIP (AT_2_ receptor interaction protein) [149]. Decreased expression of this protein results in retention of the AT_2_ receptor in cellular compartments, reducing their expression on the cell surface, and reducing AT_2_ receptor-related effects. Expression of the AT_2_ receptor at the membrane induces an increase in ATIP expression, creating a positive feedback loop [149]. PARP-1 (poly(ADP-ribose) polymerase-1) plays an important role in the regulation of AT_2_ receptor expression. Indeed, in addition to repressing the transcription of the gene coding for the AT_2_ receptor, it activates the transcription of the ATIP gene [150]. The AT_2_ receptor also interacts with another protein: the transcription factor PLZF (promyelocytic zinc finger protein). Once bound to the AT_2_ receptor, PLZF will cause its internalization, thus decreasing its membrane expression; then, PLZF will migrate to the nucleus, allowing the transcription of PI3K [151], which is involved in the activation of eNOS (endothelial nitric oxide synthase).

Although the level of expression may partly explain the predominance of the effects of one receptor over the other, it turns out that in some cases, this is more complex. For example, we have seen that after stroke, there is an increase in AT_1_ receptor-dependent vasoconstriction, despite a decrease in AT_1_ receptor gene expression [83]. Furthermore, Foulquier et al. showed that high salt intake for 4 days was associated with abolition of AT_2_ receptor-mediated vasodilation because of decreased aldosterone levels, but also with decreased cerebrovascular AT_2_ receptor protein levels without mRNA changes [152]. If this high salt intake is maintained for 30 days, in addition to the abolition of AT_2_ receptor-mediated vasodilation, we observe AT_2_ receptor-mediated vasoconstriction, although neither the mRNA nor protein levels of the AT_2_ receptor are altered by high salt intake.

### 4.2. Direct AT_1_/AT_2_ Receptors Interactions

GPCRs have the ability to form homodimers and heterodimers that can change receptor properties. In this section, we will discuss the implication of these homodimers and heterodimers on AT_1_ and AT_2_ receptor signaling (Figure 8).

#### 4.2.1. AT_1_ Receptor Dimerization

Abdallah et al. showed that increased levels of AT_1_ receptor homodimers were present on monocytes from patients with hypertension, which is an atherogenic risk factor, and that they were related to increased Ang II-dependent monocyte activity and adhesiveness [153]. This increase leads to the formation of atherosclerotic lesions. In this study, in addition to showing that inhibition of Ang II release prevents the formation of AT_1_ receptor homodimers, they observed that dimerized receptors increase G_q/11_-mediated inositol phosphate signaling [153]. In addition, the constitutive formation of homodimers of the AT_1_ receptor are formed during biosynthesis, as the receptors are trafficked through the endoplasmic reticulum. Furthermore, the constitutive nature of receptor dimerization was not affected by treatment with agonists or antagonists [154]. These data show that AT_1_ receptor homodimerization can enhance Ang II-mediated signaling, which may have a pathological effect (e.g., arthrosclerosis), but that these homodimers are not affected by receptor agonists and antagonists.

The AT_1_ receptor and B_2_ receptor can form a heterodimeric complex. This heterodimerization between the AT_1_ receptor and B_2_ receptor in HEK-293 cells increased the efficacy and potency of Ang II, but decreased the potency and the efficacy of BK. To confirm this, the authors compared the Ang II- or BK-stimulated increase in inositol phosphates in HEK-293 cells expressing the indicated receptors. Furthermore, AT_1_/B_2_ heteromerization is also involved in the increase in Ang II hypersensitivity in preeclampsia [155]. Preeclampsia is a pregnancy complication characterized by high blood pressure, and it may cause serious complications for the mother and fetus [156]. The presence of AT_1_/B_2_ heterodimers has been reported in human placental biopsies from pregnancies with preeclampsia [157]. 

In 2005, Kostenis et al. showed that the AT_1_ receptor could form a heterodimer with the MasR [158]. In their study, they transfected the human forms of Mas and AT_1_ receptors, individually and in combination, into CHO-K1 cells, and then assessed intracellular Ca^2+^ mobilization after stimulation with different doses of Ang II. The MasR alone did not respond to Ang II stimulation, while cells expressing the AT_1_ receptor increased their intracellular Ca^2+^ level. Upon coexpression of MasR with the AT_1_ receptor, a reduction in the potency and maximal efficiency of Ang II in increasing Ca^2+^ mobilization was observed. In the same model, increasing concentrations of losartan induced significant rightward shifts in Ang II concentration–response curves in cells expressing both the AT_1_ receptor alone and the AT_1_/MasR heterodimer. In contrast, the Ang II-induced elevation of intracellular Ca^2+^ and its decrease in the presence of MasR were not affected by the presence of Ang-(1-7). Furthermore, in vivo data corroborate the results obtained in cell lines, as MasR-KO animals demonstrated a significant increase in the vasoactive properties of Ang II [158]. 

#### 4.2.2. AT_2_ Receptor Dimerization

AT_2_ receptor homodimerization was first described by Miura et al. in PC12W cells and CHO cells transfected with AT_2_R [159]. They also showed that these AT_2_ receptor homodimers allow constitutive signaling that leads to apoptosis. Furthermore, dimerization and pro-apoptotic signaling were not altered following AT_2_ receptor stimulation, suggesting that this homodimerization is ligand-independent, which has also been reported in transfected HEK-293 cells [160]. Zha et al. showed in NRK-52E rat kidney epithelial cells that AT_2_ receptor homodimerization is observed under high-glucose conditions, which may be a direct effect of receptor dimerization susceptibility or an indirect effect due to increased AT_2_ receptor expression under high-glucose conditions [161]. 

Like AT_1_ receptors, AT_2_ receptors can form heterodimers with B_2_ receptors. We have already seen that AT_2_ receptors mediate a vasodilatory cascade that includes BK, NO, and cGMP. Using a KO mouse model for B_2_R, they showed that when AT_2_ and B_2_ receptors are simultaneously activated in vivo, NO and cGMP production increases [162]. In a PC12W cell model, heterodimerization of these receptors was shown without any stimulation, suggesting the presence of constitutive heterodimers [72]. Furthermore, the use of an AT_2_ receptor agonist (CGP42112A) combined with a B_2_ receptor agonist (BK) or antagonist (icatibant) allows for an increase in receptor expression alone, but also for the formation of heterodimers [72]. Thus, the maximal increase in cGMP and NO production is observed when AT_2_ receptor is stimulated and the B_2_ receptor is blocked; this result is in agreement with another study showing that B_2_ receptor blockade increases the effect of AT_2_ receptors on cGMP and NO production [163]. Since the rate of heterodimer formation depends on the expression level of AT_2_ and B_2_ receptors, controlling the expression level of these receptors would influence the formation of dimers, thus increasing (or not) the AT_2_ receptors’ effects.

The AT_2_ receptors is also able to form a heterodimeric complex with MasR [164]. This dimerization influences the RAS-protective Ang II/AT_2_ axis, resulting in increased NO production and promoting diuretic–natriuretic response in obese Zucker rats [164]. Several studies suggest that these receptors may be functionally interdependent; indeed, the AT_2_ receptor antagonist (PD123319) reduced the vasodepressor effects of the MasR agonist Ang-(1-7) [165]. Similarly, Ang-(1-7) mediated endothelium-dependent vasodilation in the cerebral arteries [166] and aortic rings of salt-fed animals [167] was inhibited by PD123319 as well as the MasR antagonist (A-779). In mouse astrocytes isolated from AT_2_ receptor KO, MasR was not responsive to Ang-(1–7), and in astrocytes isolated from MasR KO, AT_2_ receptors were not responsive to C21, suggesting that in murine astrocytes in primary culture, AT_2_ and MasR are functionally dependent on each other [168]. In contrast, another study showed that the vasoconstriction effect induced by Ang II was slightly increased, and the vasodilation induced by the AT_2_ receptor agonist CGP42112A was not altered in mice KO for MasR [169]. 

Furthermore, using bioluminescence resonance energy transfer (BRET), the AT_2_ receptor was found to form a heterodimer with the relaxin family peptide receptor 1 (RXFP1). Relaxin, by binding to RXFP1, is known to be antifibrotic by interfering with transforming growth factor β1 (TGF-β1). Ang II is also known to in-hibit TGF-β1 via the AT_2_ receptor. Chow et al. investigated the potential interactions of relaxin with the AT_2_ receptor. The antifibrotic action of relaxin in primary rat kidney myofibroblasts was reduced when combined with an AT_2_ receptor antagonist (PD123319). This heterodimerization results in the activation of the NO-cGMP-dependent pathway. Although relaxin does not interact directly with the AT_2_ receptor, AT_2_/RXFP1 heterodimerization results in the activation of the NO-cGMP-dependent pathway, leading to the disruption of TGF-β1 signaling and thus the latter’s pro-fibrotic effect.

Last but not least, AT_2_/AT_1_ heterodimerization was first described by AbdAlla et al. in PC-12 cells, rat fetal fibroblasts, and human myometrial tissue samples [170]. They showed that AT_2_/AT_1_ dimerization was constitutive and led to inhibition of the AT_1_ receptor-mediated G protein pathway. This inhibition of AT_1_ receptor signaling does not require AT_2_ receptor activation, as shown by the fact that AT_2_/AT_1_ heterodimerization is not affected by AT_2_ receptor antagonists such as PD123319, and by the persistence of the effect in cells with dimers containing an AT_2_ receptor mutant that is unable to bind agonists or to initiate AT_2_ signaling. The authors concluded that the AT_2_ receptor acts as a kind of reverse agonist of the AT_1_ receptor by constitutively preventing the conformational changes necessary to initiate AT_1_ receptor signaling [170]. 

Attenuation of AT_1_ receptor signaling by the AT_2_ receptor (via calcium signaling, ERK1/2 MAPK activation) as well as constitutive AT_2_/AT_1_ dimerization has been confirmed in studies by other groups in HeLa cells [171] or HEK-293 transfected with AT_2_/AT_1_ [172]. AT_2_/AT_1_ dimerization also appears to impact AT_2_ intracellular trafficking, as AT_2_/AT_1_ dimers internalize upon Ang II stimulation (whereas AT_2_ alone is unable to do so) [171].

More recently, a study demonstrated that AT_1_ and AT_2_ receptors form heterodimers that are expressed in the cells of the central nervous system (striatal neurons and microglia). These dimers are new functional units with specific signaling properties, because on the one hand, coactivation of the two receptors reduces Ang II signaling, and on the other hand, they exhibit cross-potentiation: i.e., candesartan (AT_1_ receptor antagonist), increases the effect of AT_2_ receptor agonists [172].

In a model of Parkinson’s disease (6-OH-dopamine hemi-lesioned rat), the authors wanted to quantify the quantity of AT_2_/AT_1_ dimers in striatal sections of naïve and 6-OH-dopamine hemi-lesioned rats, treated or not with L-DOPA and divided into two groups: those that are dyskinetic and those that are resistant to L-DOPA-induced dyskinesia. First, they demonstrated that the quantity of AT_2_/AT_1_ dimers found in the non-lesioned striatum was negligible compared to the lesioned striatum. Furthermore, dyskinetic animals on L-DOPA showed an approximately 2-fold increase in AT_2_/AT_1_ dimers (compared to the lesioned rat hemisphere), and dyskinesia-resistant animals showed an approximately 10-fold increase (compared to the non-lesioned control hemisphere) [172]. In this context of Parkinson’s disease, the use of AT_1_ receptor antagonists coupled to AT_2_ receptor agonists could potentiate the neuroprotective effects via the AT_2_ receptor.

**Figure 8 molecules-28-05481-f008:**
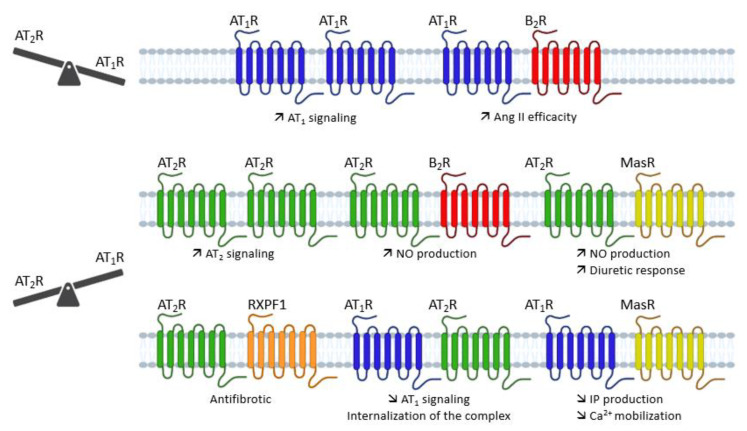
Overview of the different dimerizations of AT_1_ and AT_2_ receptors and their functional impact. Ang II: angiotensin II; AT_1_R: angiotensin type 1 receptor; AT_2_R: angiotensin type 2 receptor; B_2_R: bradikynin receptor B2; Ca^2+^: calcium; IP: inositol phosphate; MasR: Mas receptor; NO: nitric oxide; RXPF1: relaxin family peptide receptor 1.

In conclusion, the different dimerizations of AT_1_ or AT_2_ receptors, depending on the tissue and the co-expressed receptor (AT_1_, AT_2_, B_2_R, MasR), cause a functional change in the activity of the receptors, modifying the AT_1_/AT_2_ balance without changing the expression of these receptors.

### 4.3. Post-Translational Modifications

Post-translational modifications are also mechanisms to modulate GPCR functions, AT_1_ and AT_2_ receptors being no exception.

#### 4.3.1. *N*-Glycosylation

AT_1_ receptor function is regulated by various post-translational modifications such as *N*-glycosylation [173]. To study the effects of *N*-glycosylation on extracellular loops (ECL), artificial *N*-glycosylation sequences were incorporated into ECL1, ECL2 and ECL3 [174]. In ECL1, *N*-glycosylation causes a very significant decrease in the ligand affinity and surface expression of the receptor; in ECL2, it leads to the synthesis of a misfolded receptor, and in ECL3 *N*-glycosylation produces mutant receptors with normal affinity and low surface expression. These results show that *N*-glycosylation sites alter many properties of the AT_1_ receptor, such as targeting, folding, affinity, and surface expression [174].

Like the AT_1_ receptor, the AT_2_ receptor contains multiple glycosylation sites. The variability in AT_2_ receptor molecular weight has been shown to be due to different degrees of *N*-glycosylation [175]. However, this glycosylation does not appear to be involved in AT_2_ receptor binding and membrane addressing [176]. On the other hand, glycosylation of AT_2_ receptor could nevertheless play a role in the stability of the receptor and its coupling with its intracellular effectors [176].

#### 4.3.2. Phosphorylation

Phosphorylation is another post-translational modification that can occur on the AT_1_ receptor [177]. Phosphorylation of AT_1_ receptor serine/threonine residues is required for β-arrestin recruitment and receptor internalization [178]. In contrast, in the kidneys and arteries, GRK4 has been shown to exacerbate urinary sodium retention and vasoconstriction by increasing AT_1_ receptor expression [179]. 

The AT_2_ receptor is rapidly phosphorylated on a serine by PKC after activation by Ang II. The functional role of AT_2_ receptor phosphorylation is not known. When triggered by AT_1_ receptor activation, it appears to modulate the AT_2_ receptor’s effects, as opposed to AT_1_’s effects [180]. 

A more recent study showed that GRK4, by increasing AT_2_ receptor phosphorylation, impairs AT_2_ receptor-mediated diuresis and natriuresis without decreasing mRNA or protein levels [181]

#### 4.3.3. *S*-Nitrosation

*S*-nitrosation is a mode of post-translational modification that allows the addition of a NO group to the sulfur atom of specific cysteine residues. 

The AT_1_ receptor contains ten cysteine residues. Four of these cysteines are involved in disulfide bridges at the extracellular loops, one is located on the cytoplasmic tail, and the other five are distributed in the transmembrane domains of the receptor (see Section 2.1.1). The affinity of the AT_1_ receptor for Ang II is decreased in the presence of sodium nitroprusside (SNP), an NO donor. An assessment of the affinity of different mutated AT_1_ receptors for each of five cysteines revealed the cysteine involved in sensitivity to sodium nitroprusside, cysteine residue 289 [182].

The AT_2_ receptor has 14 cysteine residues. We have seen that the four cysteines in the extracellular loops are engaged in disulfide bridges. The remaining ten cysteines are distributed on the transmembrane domains and the cytoplasmic tail.

In 2015, Jang showed that the AT_1_ receptor, AT_2_ receptor, and ROS appear to be involved in increasing nNOS (neuronal nitric oxide synthase) activity [183]. After stimulation with Ang II, membrane expression of the AT_1_ receptor decreases, while that of the AT_2_ receptor increases. In addition, losartan and L-NAME (eNOS inhibitor) inhibit translocation to the membrane of AT_2_ receptor, suggesting that AT_2_ receptor translocation may be NO-dependent. The SNP allows for the *S*-nitrosation of the AT_2_ receptor, as it does for the AT_1_ receptor. In addition, a mutation in cysteine 349 induces increased surface expression of the AT_2_ receptor, suggesting that it plays a role in the translocation of the receptor to the membrane [183].

## 5. Possible Ways to Tune the AT_1_/AT_2_ Functional Balance

### 5.1. Pushing the Balance Using Agonist/Antagonist Ligands

One of the most obvious means to act on the AT_1_/AT_2_ functional balance is to use agonists or antagonists towards one or the other receptor.

In order to rebalance the AT_1_/AT_2_ balance, the use of specific molecules targeting our receptors seems to be an interesting avenue. Indeed, this would make it possible to block or activate one specific receptor, which cannot be achieved with ACE inhibitors, for example, as they prevent both receptors’ activation by inhibiting Ang I cleavage. 

AT_1_ receptor antagonists were the first molecules discovered in this sense (Table 1). In the case of pathologies associated with AT_1_ receptor, the ideal is to be able to abolish the overexpression of the receptor responsible for the imbalance in order to orient it in favor of the AT_2_ receptor. AT_1_ receptor antagonists are widely used to treat hypertension as well as cardiac diseases (heart failure or myocardial infarction). In vitro and in vivo, losartan is a reversible competitive AT_1_ receptor antagonist that inhibits the Ang II-induced vasoconstriction of blood [184]. However, complete blockade of the receptor then amounts to promoting activation of the AT_2_ receptor, which will tip the balance in the other direction instead of bringing it back to equilibrium.

Another method would be to act on the AT_2_ receptor itself by using an agonist of the latter. For this, molecules capable of specifically activating the receptor have been developed, such as CGP42112A, which is a peptide, or C21, which is a synthetic compound. A study on the effects of CGP42112A in the same SHR model was performed, and the authors tested CGP42112A in the presence or absence of candesartan. The results showed that the use of candesartan alone at a high concentration lowered blood pressure in SHR rats, and that CGP42112A only provided a depressant effect in the presence of candesartan [185]. Another study on CGP42112A showed that intravenous infusion of the molecule increased NO production in pigs [186]. Wan et al. showed that the use of C21 in hypertensive rats reduced blood pressure [187]; however Foulquier et al. showed that this reduction in blood pressure was dependent on the experimental model used [77].

These studies show that the use of an AT_2_ receptor agonist alone would not be sufficient to regulate the balance given the predominance of the AT_1_ receptor in the pathological context. However, it would appear that combining the effects of an AT_1_ receptor antagonist and an AT_2_ receptor agonist would be more effective in certain pathologies such as hypertension [188,189].

We have seen that the AT_2_ receptor has a rather protective role in case of pathology, but a study has shown that the use of PD123319 can reduce the inflammatory response and oxidative stress in rats with induced colitis [190]. In this same study and another one, it was demonstrated that PD123319 may have partial agonistic properties in relation to the AT_2_ receptor, thus biasing the interpretation of functional data [190,191]. Another study conducted by our group showed that in an SHR model, the combination of the effects of an ARB with an ACE inhibitor induced an exacerbated decrease in middle cerebral artery contraction in SHR [192]. In this specific case, by blocking the AT_1_ receptor, the AT_2_ receptor will become dominant, thereby reversing the balance. This phenomenon has been described by Budzyn et al. [193].

**Table 1 molecules-28-05481-t001:** Angiotensin receptor ligands.

Agonists			
Compounds	AT_1_ affinity	AT_2_ affinity	References
Ang II	pIC_50_ = 8.1	pIC_50_ = 9.2	[194]
Ang III	pIC_50_ = 7.6	pIC_50_ = 9.2	[194]
Ang IV	N.A.	pIC_50_ = 7.3	[194]
Ang-(1-7)	pKi = 6.66	pIC_50_ = 6.6	[194,195]
SII	pKd = 6.5	N.A.	[196]
TRV120023	pEC_50_ = 7.4	N.A.	[197]
TRV120026	pEC_50_ = 7.6	N.A.	[197]
TRV120027	pEC_50_ = 7.7	Ki = 7 nM	[197]
CGP42112A	N.A.	pIC_50_ = 9.6	[194]
C21	N.A.	pIC_50_ = 8.6	[194]
Antagonists			
Compounds	AT_1_ affinity	AT_2_ affinity	References
Losartan	pIC_50_ = 7.4–8.7	N.A.	[198]
Candesartan	pIC_50_ = 9.5–9.7	N.A.	[199]
Valsartan	pIC_50_ = 8.6	N.A.	[200]
Telmisartan	pIC_50_ = 8.4	N.A.	[201]
PD123177	N.A.	pIC_50_ = 8.5–9.5	[202]
PD123319	N.A.	pIC_50_ = 8.25	[194]

N.A.: non-applicable. EC_50_: half maximum effective concentration; IC_50_: half maximum inhibitory concentration; Kd: dissociation constant. Ki: inhibition constant.

### 5.2. Selective Activation of the β-Arrestin Pathway

The use of biased agonists is another strategy to regulate the AT_1_/AT_2_ receptors’ balance. A biased agonist is a receptor-specific ligand capable of selectively activating a single signaling pathway by preferentially stabilizing one of the receptor conformations. This phenomenon is also called “functional selectivity”. In the case of the AT_1_ receptor, several biased agonists have been developed that allow AT_1_ receptor to adopt an alternative active conformation [45]. For example SII, TRV120027, and TRV120023 selectively activate the β-arrestin pathway while inhibiting the G protein pathway [197]. 

SII is a modified Ang II peptide, which can trigger the phosphorylation of the AT_1_ receptor, and thus β-arrestin recruitment [203]. SII elicits GRK6 and β-arrestin 2-dependent ERK activation, and promotes β-arrestin-regulated Akt activity and mTOR phosphorylation to stimulate protein synthesis [204]. TRV120023 has been reported to only recruit β-arrestin while blocking G protein activation, enhancing myocyte contractility but without promoting hypertrophy, as seen with Ang II [205]. For example, an acute infusion of Ang II increased mean arterial pressure in male SHR, accompanied by a reduction in glomerular filtration rate, whereas TRV120023 blocked the acute infusion of an Ang II-induced hypertensive state in a dose-dependent manner [205]. 

TRV120027 has been studied in several preclinical studies in a heart failure model, and has shown promising results [206]. A recent study just demonstrated that β-arrestin signaling, mediated by the PAR-1 receptor, produces prolonged activation of MAPK 42/44, which increases PDGF-β secretion. It has been shown that after ischemic stroke, PDGF-β secretion can provide increased protection of endothelial function and barrier integrity [207]. Similarly, there are biased agonists that selectively activate the G protein pathway, such as TRV120055 or TRV120056 [33].

The use of biased agonists could be interesting in a pathological setting. Indeed, unlike ARBs, which completely inhibit the effects of the AT_1_ receptor, β-arrestin-biased agonists will allow the activation of the β-arrestin pathway, with potentially protective effects. Activation of the β-arrestin pathway to mediate internalization could furthermore reduce the number of AT_1_ receptors present at the surface. On the other hand, if AT_1_ receptors are already occupied by a biased agonist, Ang II could then be able to bind to the AT_2_ receptor. These combined effects could restore the functional balance without blocking the AT_1_ receptor. In addition to this internalization, β-arrestin may activate beneficial secondary signaling pathways [207].

Although TRV120027 is able to bind to the AT_2_ receptor with an affinity comparable to that of the AT_1_ receptor [197], no functional studies have been carried out. Furthermore, as the receptor is unable to interact with β-arrestin and be internalized, most signaling pathways function through the stimulation of the G protein pathway.

### 5.3. Post-Translational Regulation

We have previously shown that AT_1_ and AT_2_ receptors can undergo different post-translation changes (see Section 4.3). 

Regarding phosphorylation, we have already established that the AT_2_ receptor was not able to recruit β-arrestins as would classically be the case for GPCRs, including the AT_1_ receptor. On the other hand, intervention on certain GRKs, such as GRK4, could have an effect on the AT_1_/AT_2_ balance because it would allow, on the one hand, the inhibition of the overexpression of AT_1_ receptors at the membrane, and on the other hand, a decrease in the phosphorylation of the AT_2_ receptor and thus the restoration of AT_2_ receptor signaling.

Although the glycosylations of the AT_1_ receptor are very important for its membrane addressing and ligand binding, for the AT_2_ receptor, the role of these glycosylations remains unknown. 

AT_1_ and AT_2_ receptors both have four cysteines involved in disulfide bridges. It has been shown that reduction of these disulfide bridges by dithiothreitol (DTT) strongly decreases Ang II binding for the AT_1_ receptor, but not for the AT_2_ receptor [32].

AT_1_ and AT_2_ receptors can undergo *S*-nitrosation. In a study of our group, rat middle cerebral arteries were pretreated with an NO donor, *S*-nitrosoglutathione (GSNO), and this led to abolition of Ang II-induced vasoconstriction [208]. The following results were found in an ex vivo rat aortic ring model. A decrease in Ang II-induced vascular contraction was observed when arteries were pretreated with GSNO. In vivo, a decrease in blood pressure response to Ang II was also detected after oral administration of GSNO [209]. In vitro experiments showed that pretreatment of HEK293 cells with GSNO seems not to alter the internalization of activated AT_1_ receptors [208]. Similar to biased agonists, S-nitrosation disrupts AT_1_ receptor G protein signaling, the advantage here being that NO is an endogenous product.

This post-translational modification seems to be an interesting tool for directing a rather protective response by not altering the AT_1_/AT_2_ balance as antagonists and/or agonists can. Indeed, we have seen that NO is one of the main second messengers of AT_2_ receptors. We can envisage that the NO produced by the AT_2_ receptor, which is even more important when the AT_2_ receptor and B_2_ receptor form a dimer, could *S*-nitrosate the AT_1_ receptor. 

In general, the use of biased agonists of the β-arrestin pathway and *S*-nitrosation of the AT_1_ receptor allow for the interruption of the G-protein pathway. In addition, biased agonists allow the activation of signaling pathways with protective effects similar to those of the AT_2_ receptor.

## 6. Conclusions

In summary, we have seen through several examples that the AT_1_/AT_2_ balance is very important from a physiological point of view, and that the disturbance of this balance leads to the appearance of pathologies, most often as a result of the dominance of the AT_1_ receptor over the AT_2_ receptor. However, emerging studies have shown that secondary signaling of β-arrestins could have beneficial effects. This is why a finer regulation of this balance via *S*-nitrosation of the receptors or the use of biased agonists seems to be interesting, and could open new therapeutic perspectives.

## Figures and Tables

**Figure 1 molecules-28-05481-f001:**
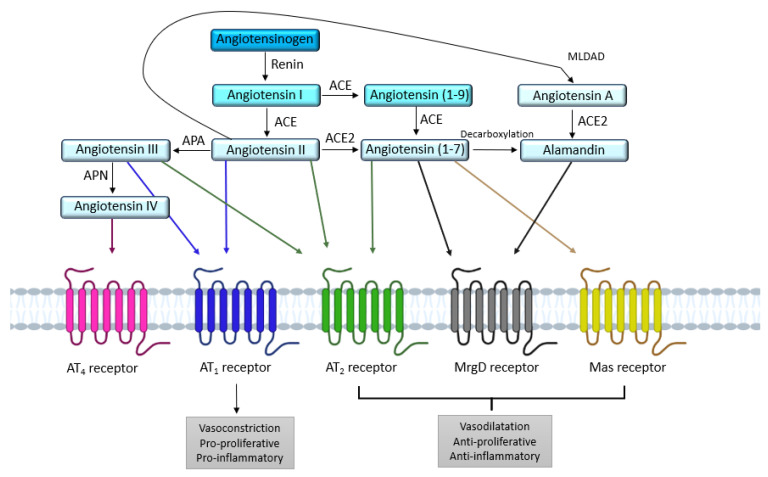
Overview of the different receptors and enzymes involved in the renin angiotensin system. ACE: angiotensin I-converting enzyme; ACE2: angiotensin II-converting enzyme 2; APA: aminopeptidase A; APN: aminopeptidase N; AT_1_: angiotensin II type 1 receptor; AT_2_: angiotensin II type 2 receptor; MLDAD: mononuclear leukocyte-derived aspartate decarboxylase; MrgD: Mas-related G protein-coupled receptor, member D.

**Figure 6 molecules-28-05481-f006:**
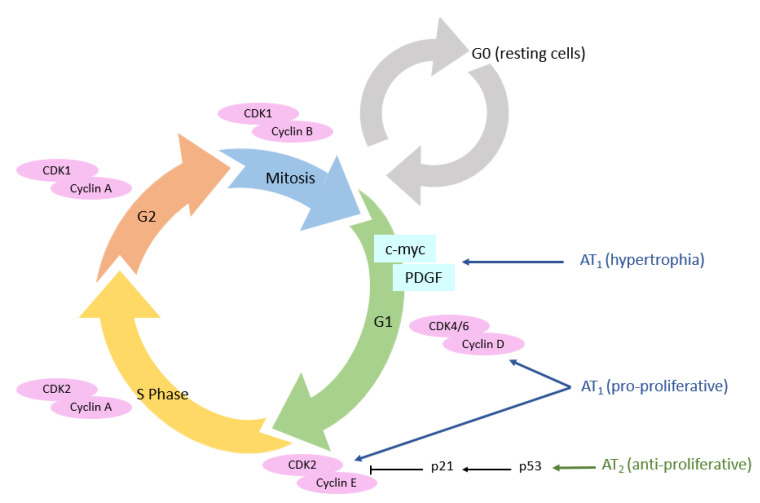
AT_1_ and AT_2_ receptors’ expression and their impact on cell cycle. AT_1_ receptor: angiotensin II type 1 receptor; AT_2_ receptor: angiotensin II type 2 receptor; CDK: cyclin-dependent kinases.

**Figure 7 molecules-28-05481-f007:**
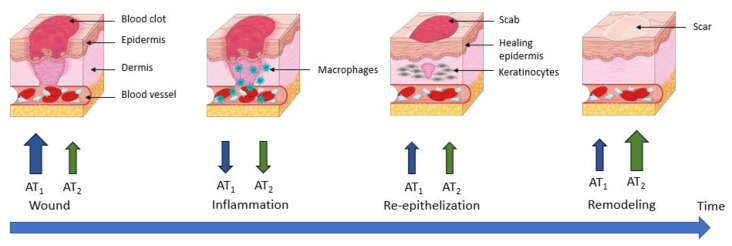
AT_1_ and AT_2_ receptors’ expression during wound healing. AT_1_ receptor: angiotensin II type 1 receptor; AT_2_ receptor: angiotensin II type 2 receptor.

## Data Availability

No new data were created or analyzed in this study. Data sharing is not applicable to this article.

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
