# Peer review of "The AT1/AT2 Receptor Equilibrium Is a Cornerstone of the Regulation of the Renin Angiotensin System beyond the Cardiovascular System"

_molecules, 2023, doi:10.3390/molecules28145481_

Round 1
Reviewer 1 Report
I really enjoyed reading this review and I think it provides a point of difference to the many reviews in this space. Well done. I have a few comments which I think will elevate the paper;
It is universally accepted among experts in the field that there is no AT3 receptor. I would remove this from the manuscript.
The AT4 receptor is an enzyme, also known as IRAP. Some Angiotensin peptides do bind/inhibit this enzyme which causes effects similar to AT2 activation. I think this deserves a mention at least in the introduction, particularly since IRAP is involved in the synthesis of its own substrate.
The MRGD receptor is also a member of the RAS family of receptors. Some mention of it should also be present in the introduction. Ideally, the introduction should feature a schematic. There are many receptors/enzymes/peptides involved so highlighting the key players would set the scene.
I would also have a schematic of receptor signalling for AT1 and AT2, side-by-side, just to highlight the key differences between the receptors.
In terms of cancer, AT2R is a controversial topic and there is certainly conflicting literature. I therefore have a problem with the following statement
"It seems that this overexpression of AT2 would counterbalance that of AT1 in order to prevent the proliferation of cancer cells and to induce their apoptosis"
There is evidence to the contrary, most recently by Perryman (PNAS, 2022). I would recommend to either remove the above comment or temper the language to suggest that there is still some way to go before we understand the role of AT2 in cancer.
There is evidence for the formation of functional RXFP1:AT2 dimers. They should also be mentioned within the heterodimerisation section.
The paper would benefit from English editing as there are a number of grammatical or spelling issues throughout the paper.
Author Response
I really enjoyed reading this review and I think it provides a point of difference to the many reviews in this space. Well done. I have a few comments which I think will elevate the paper;
- It is universally accepted among experts in the field that there is no AT3 receptor. I would remove this from the manuscript.
We agree with the referee and removed this sentence accordingly.
- The AT4 receptor is an enzyme, also known as IRAP. Some Angiotensin peptides do bind/inhibit this enzyme which causes effects similar to AT2 activation. I think this deserves a mention at least in the introduction, particularly since IRAP is involved in the synthesis of its own substrate.
We agree with the referee and added a section in the introduction to clarify this point:
Please see page 2, lines 53-57: “The AT4 receptor, whose ligand is Ang IV, has been identified as a transmembrane enzyme, insulin-regulated membrane aminopeptidase (IRAP) [14]. In addition to its vasorelaxant effect in cerebral [15] and renal [16] vascular beds, the AT4 receptor seems to be involved in memory and in Alzheimer's disease [17,18].”
- The MRGD receptor is also a member of the RAS family of receptors. Some mention of it should also be present in the introduction. Ideally, the introduction should feature a schematic. There are many receptors/enzymes/peptides involved so highlighting the key players would set the scene.
We added a phrase in the introduction to cite the MrgD receptor as well as a figure to present the main receptors and enzymes involved in the RAS
Please see page 2, lines 51-52: “In addition, decarboxylation of Ang(1-7) transforms it into alamandin, which is also able to bind to the MrGD receptor.”
Please see Figure 1 page 2
- I would also have a schematic of receptor signalling for AT1 and AT2, side-by-side, just to highlight the key differences between the receptors.
We added 2 figures in the new version of the manuscript to highlight the most important signaling cascades for both receptors. However, we did not place them side-by-side, as suggested by the reviewer, as we believe it would be clearer for the reader to have these figures within the sections presenting each receptor. We can still change for a side-by-side figure at the reviewer’s will.
Please see Figure 3 page 6 for AT1 and Figure 5 page 8 for AT2.
- In terms of cancer, AT2R is a controversial topic and there is certainly conflicting literature. I therefore have a problem with the following statement
"It seems that this overexpression of AT2 would counterbalance that of AT1 in order to prevent the proliferation of cancer cells and to induce their apoptosis"
There is evidence to the contrary, most recently by Perryman (PNAS, 2022). I would recommend to either remove the above comment or temper the language to suggest that there is still some way to go before we understand the role of AT2 in cancer.
We agree with the referee and removed this statement from the new version of the manuscript.
- There is evidence for the formation of functional RXFP1:AT2 dimers. They should also be mentioned within the heterodimerisation section.
We thank the reviewer for his comment and added a new section to describe this RXFP1:AT2 dimer. It has also been added in Figure 8.
Please see page 17, lines 745-755: “Furthermore, using […] pro-fibrotic effect.”
Please see Figure 8 page 18
Reviewer 2 Report
In the article “The AT1/AT2 receptor equilibrium is a cornerstone for the regu-2 lation of the renin angiotensin system”, Colin et al review the properties of angiotensin receptors, specifically the effect of these receptors on circulation as well as other less well-known effects of the angiotensin system. The review provides several examples, and highlights the interaction of AT1R and AT2 receptors in the physiological role of Angiotensin at a systemic as well as at local level.
Although the review is
1) The introduction starts with a paragraph emphasizing the importance of a research group in the field of angiotensin regulation of cerebral circulation. The importance of this group is not well described, and is based on the inclusion of 7 citations, some of which are later referenced throughout the article. I would recommend use another section (maybe 3.2?) to highlight their work, and unless it is justified, it should be eliminated altogether (to avoid inappropriate self-citations).
2) The review is focused on the effect of RAS on the vasculature. It also includes interesting examples of the angiotensin in cell cycle, inflammation and wound healing. Although the choice of the targets of the RAS is not objectionable and was reflected in the abstract, it was not included in the title, that has to be modified to clarify this, and indicated in the body of the article that it is not a systematic review, and if possible why the authors have chosen those examples.
3) In line 287, the authors indicated that the number of these examples will be three. This is confusing, since later there are 4 sections (3.1 to 3.4)
4) When describing “Post-translation interaction” (section 4,3), most of them are described for each of the receptors, and little interaction between the two receptors (mofified or not) is described? Please, consider if this should be placed when describing the structure of the receptors. Also, section 5.3 Post-translational regulation could be grouped with these?
Minor points
I would suggest to include some illustration of the different pathways reflecting the interaction of the AT1 and AT2 receptors, since it is one of the focus of the review.
There are several minor issues with the text in the opinion of this reviewer. Please kindly consider these and ignore those that are not relevant. But seem to me that there are too many to ignore them:
In the structure of the AT2 receptors (lines 216), it is said “In contrast to the AT1 receptor, the first crystalline structures of the AT2 receptor were 216 published in 2017 [55]” What is in contrast to the AT1 receptor? The year of publication? This seems confusing to me
In line 339, the authors wrote: “The two receptors are expressed inside or near the brain cardiovascular centers (the 339 medulla oblongata).” Is the medulla oblongata the brain cardiovascular centers? Later several nuclei are mentioned, and again a similar expressions are used later in the article (“in these cardiovascular brain regions” Lines 348-349, “the neurons of the cardiovascular centers”, lines 361-362 ). This seems confusing to me
The authors wrote in lines 369-370 that “Furthermore, it has been shown that AT2 receptor expression is increased in the MVR and SNT of hypertensive rodents, resulting in a decrease in blood pressure.” Not sure if I understand this sentence, the blood pressure is increased (hypertensive rodents) or reduced (resulting in a decrease in blood pressure).
The last sentence in the first paragraph of page 9 (lines 377-381) is too long and contains several words with similar meaning, might be useful to simplify it. Also, check the wording, are you sure you wanted to write “cytokine-producing”.
In line 407 the authors wrote “The proliferation of normal cells is regulated by the cell cycle”. This is confusing to me, as I consider the cell cycle as the events that cause cell growth and division itself, and not the regulation. The regulation is done by kinases (CDKs), that are themselves regulated by cyclins, and these cyclins are not subunits of the CDK. Please check and correct where necessary.
In lines 793-794 the authors indicate that “which is not the case of ACE inhibitors for example, since they block both receptors.” Do they?
In line 798. “They” refers to the AT1 receptor antagonists? These are not mentioned in the previous sentence, but the one before.
In line 822, the authors wrote “In this same study and another one…”, but finish the sentence with two references [192,193], suggesting that one of the references should have been [184], or that there are two additional studies besides [184]
In line 869, the authors wrote “there are no functional studies existing [191].” Not sure it that is the right wording. Also, consider placing the reference before this sentence, with the previous sentence.
Formatting errors: The text contains letters in different fonts. Besides references, other examples can be found in line 262 or line 244 (font size?). In line 268, paragraph ends without dot. These should be revised and corrected
Grammatical errors: In line 340, is “There are” correctly used? In line 342, “These studies” refer to one study, [31]? Would it be possible to use more examples for determination of localization of AT1 receptor by autoradiography?
In line s 611-612, subject (“formation of homodimers” and the verb, “are”, are separated by a comma.
Spelling errors: Please, check the following and correct if necessary_
Line 448: “In esophageal adenocarinomal cell (EAC)”
CGP42112A is spelled several times as CPG42112A, specially in the paragraph in lines 804-814, where the two formats are used.
Would it be better to eliminate the section Acknowledgments? Seems to be not edited from a template
References include the month of the year in French?
Previously commented
Author Response
In the article “The AT1/AT2 receptor equilibrium is a cornerstone for the regulation of the renin angiotensin system”, Colin et al review the properties of angiotensin receptors, specifically the effect of these receptors on circulation as well as other less well-known effects of the angiotensin system. The review provides several examples, and highlights the interaction of AT1R and AT2 receptors in the physiological role of Angiotensin at a systemic as well as at local level.
Although the review is
- The introduction starts with a paragraph emphasizing the importance of a research group in the field of angiotensin regulation of cerebral circulation. The importance of this group is not well described, and is based on the inclusion of 7 citations, some of which are later referenced throughout the article. I would recommend use another section (maybe 3.2?) to highlight their work, and unless it is justified, it should be eliminated altogether (to avoid inappropriate self-citations).
As the present manuscript is submitted to a special issue devoted to the 150th anniversary of the school of pharmacy of Nancy (France), we decided to open this review by a tribute to Pr J. Atkinson (1943-2023) who has just passed away. Pr J. Atkinson has been teaching Pharmacology at the school of pharmacy of Nancy from 1986 till his retirement in 2012.
We would thus rather keep this short tribute to introduce the paper. However, we agree with the reviewer about his concern related to self-citations and removed the references in which authors of the present paper were co-authors and rewrote this section to clarify with tribute to Pr Jeffrey Atkinson.
Please see page 1, lines 30-34: “Over the past decades, the team of Professor Jeffrey Atkinson, who recently passed away (1943-2023) and who teached Pharmacology at the Faculty of Pharmacy of Nancy for over 25 years, has contributed to demonstrate the major role of the renin angiotensin system (RAS) on the regulation of the cardiovascular system and the cere-bral circulation [1-4].”
- The review is focused on the effect of RAS on the vasculature. It also includes interesting examples of the angiotensin in cell cycle, inflammation and wound healing. Although the choice of the targets of the RAS is not objectionable and was reflected in the abstract, it was not included in the title, that has to be modified to clarify this, and indicated in the body of the article that it is not a systematic review, and if possible why the authors have chosen those examples.
We agree with the reviewer and we slightly changed the title of the manuscript to illustrate that this review does not only focus on the vasculature and cardiovascular system. We also added a sentence to clarify the fact that the present paper is not a systematic review.
Please see page 3, lines 104-106: “We will not illustrate this point by a systematic review but through several examples chosen to emphasize the ubiquitous aspect of this major regulation of physiological functions.”
- In line 287, the authors indicated that the number of these examples will be three. This is confusing, since later there are 4 sections (3.1 to 3.4).
We apologize for this confusion and corrected the sentence accordingly.
Please see page 9, lines 344-346: “We will now discuss, through four different examples, the physiological and pathophysiological implication of the AT1/AT2 functional balance [72].”
- When describing “Post-translation interaction” (section 4,3), most of them are described for each of the receptors, and little interaction between the two receptors (modified or not) is described? Please, consider if this should be placed when describing the structure of the receptors. Also, section 5.3 Post-translational regulation could be grouped with these?
We agree with the reviewer that the title of section 4.3 may be confusing as it does not deal about “interaction” between AT1 and AT2. The title of section 4.3 has been changed for “4.3 Post-translational modifications” (see page 18 line 800).
However, in our opinion, this section should stay in chapter 4 as well as section 5.3 in chapter 5. Our idea is to only describe the structures of the receptors in chapter 2 (this is why we just emphasize on the different sites on the structure of the receptor that can be modified). Chapter 4 describes the acknowledged functions of these modifications while Chapter 5 gives a few perspectives on how one can act on these modifications to regulate the functions of these receptors.
As the other reviewer did not criticize the overall organization of the manuscript, we chose to keep it like this. We may reconsider this at the reviewer’s will.
Minor points
a) I would suggest to include some illustration of the different pathways reflecting the interaction of the AT1 and AT2 receptors, since it is one of the focus of the review.
We added three figures according to reviewer #1’s comment: one in the introduction showing the different receptors and ligands involved in the RAS (Figure 1, page 2), and the two others to describe the signaling pathways for AT1 (Figure 3, page 6) and AT2 (Figure 5, page 8) receptors.
Illustrating in one single figure the different levels (functional, post-translational, signaling pathways…) of the AT1/AT2 interactions appears difficult and our attempts were not satisfying: the figures are unreadable and confusing. We believe that the added figures will help the reader to better understand the different crosstalk between AT1 and AT2.
b) There are several minor issues with the text in the opinion of this reviewer. Please kindly consider these and ignore those that are not relevant. But seem to me that there are too many to ignore them:
In the structure of the AT2 receptors (lines 216), it is said “In contrast to the AT1 receptor, the first crystalline structures of the AT2 receptor were 216 published in 2017 [55]” What is in contrast to the AT1 receptor? The year of publication? This seems confusing to me.
We agree with the reviewer and removed this assessment from the sentence.
Please see page 6, lines 270-272: “The first crystalline structures of the AT2 receptor were published in 2017 [54] demonstrating commonalities such as an extracellular loop 2 (ECL2) β-hairpin conformation.”
c) In line 339, the authors wrote: “The two receptors are expressed inside or near the brain cardiovascular centers (the 339 medulla oblongata).” Is the medulla oblongata the brain cardiovascular centers? Later several nuclei are mentioned, and again a similar expressions are used later in the article (“in these cardiovascular brain regions” Lines 348-349, “the neurons of the cardiovascular centers”, lines 361-362 ). This seems confusing to me
We agree with the reviewer that this section is confusing. All the brain centers we cite are involved in cardiovascular regulation. We rewrote this section to clarify this point.
Please see page 10, lines 400-408: “The two receptors are expressed […] cardiovascular control centers [85].
d) The authors wrote in lines 369-370 that “Furthermore, it has been shown that AT2 receptor expression is increased in the MVR and SNT of hypertensive rodents, resulting in a decrease in blood pressure.” Not sure if I understand this sentence, the blood pressure is increased (hypertensive rodents) or reduced (resulting in a decrease in blood pressure).
Indeed, the sentence was wrong and confusing. It has been corrected in the new version of the manuscript.
Please see page 10, lines 428-429: “Furthermore, the stimulation of RVLM AT2 receptor by a specific agonist (CGP42112A) results in a drop in blood pressure [90].”
e) The last sentence in the first paragraph of page 9 (lines 377-381) is too long and contains several words with similar meaning, might be useful to simplify it. Also, check the wording, are you sure you wanted to write “cytokine-producing”.
We agree with the reviewer and changed this paragraph accordingly.
Please see page 10-11, lines 435-437: “In general, the activation of the AT1 receptor in macrophages leads to the activation of the pro-inflammatory axis of the RAS while activation of the AT2 receptor promotes the activation of an anti-inflammatory axis [91,92].”
f) In line 407 the authors wrote “The proliferation of normal cells is regulated by the cell cycle”. This is confusing to me, as I consider the cell cycle as the events that cause cell growth and division itself, and not the regulation. The regulation is done by kinases (CDKs), that are themselves regulated by cyclins, and these cyclins are not subunits of the CDK. Please check and correct where necessary.
We agree with the reviewer and changed this sentence to clarify.
Please see page 11, lines 486-491: “The proliferation of normal cells is regulated by kinases called cyclin-dependent kinases (CDKs). The main actors in this cell cycle are cyclins, which regulate CDKs, enabling cells to progress through the cell cycle.[96]”
g) In lines 793-794 the authors indicate that “which is not the case of ACE inhibitors for example, since they block both receptors.” Do they?
The reviewer is right, as the phrasing was confusing. We changed the sentence to correct this.
Please see page 19, lines 863-865: “Indeed, this would make it possible to block or activate specifically one receptor, which cannot be obtained with ACE inhibitors, for example, as they prevent both receptors activation by inhibiting Ang I cleavage.”
h) In line 798. “They” refers to the AT1 receptor antagonists? These are not mentioned in the previous sentence, but the one before.
We replaced ‘They” by “AT1 receptor antagonists” (Please see page 19, line 869)
i) In line 822, the authors wrote “In this same study and another one…”, but finish the sentence with two references [192,193], suggesting that one of the references should have been [184], or that there are two additional studies besides [184]
Indeed, the numbering of the references was not right and has been corrected.
j) In line 869, the authors wrote “there are no functional studies existing [191].” Not sure it that is the right wording. Also, consider placing the reference before this sentence, with the previous sentence.
We have changed the sentence according to the reviewer’s suggestion.
Please see page 21, lines 940-941: “Although TRV120027 is able to bind to the AT2 receptor with an affinity comparable to that of the AT1 receptor [194], no functional studies have been carried out.”
k) Formatting errors: The text contains letters in different fonts. Besides references, other examples can be found in line 262 or line 244 (font size?). In line 268, paragraph ends without dot. These should be revised and corrected.
The formatting errors were revised.
l) Grammatical errors: In line 340, is “There are” correctly used? In line 342, “These studies” refer to one study, [31]? Would it be possible to use more examples for determination of localization of AT1 receptor by autoradiography?
We corrected “there are” for “they are” and added 2 other references using autoradiography to localize AT1 and AT2 receptors.
Please see page 10, lines 397-401: “They are exclusively found in the neurons rather than the glia. The localization of AT1 receptors in the brain has been determined primarily by receptor autoradiography [32,84,85]. These studies demonstrated a wide […]”
m) In lines 611-612, subject (“formation of homodimers” and the verb, “are”, are separated by a comma.
The comma was removed.
Please see page 15, lines 670-672: “In addition, the constitutive formation of homodimers of AT1 receptor are formed during biosynthesis as the receptors trafficked through the endoplasmic reticulum.”
n) Spelling errors: Please, check the following and correct if necessary:
Line 448: “In esophageal adenocarinomal cell (EAC)”
This has been corrected.
Please see page 12, lines 506-507: “In esophageal adenocarcinomal cell (EAC) Fujihara et al., showed that telmisartan induces antitumoral effects in EAC, both in vitro and in vivo.”
CGP42112A is spelled several times as CPG42112A, specially in the paragraph in lines 804-814, where the two formats are used.
We corrected and homogenized throughout the manuscript.
Please see page 16, lines 720-723: “Furthermore, the use of an AT2 receptor agonist (CGP42112A) combined with a B2 receptor agonist (BK) or antagonist (icatibant) allows for an increase in re-ceptor expression alone but also for the formation of heterodimers [69].”
Please see page 20, lines 876-882: “For this, molecules capable of specifically activating the receptor have been developed, such as CGP42112A which is a peptide or C21 which is a synthetic com-pound. A study on the effects of CGP42112A in the same SHR model was performed, the authors tested CGP42112A in the presence or absence of candesartan. The results showed that the use of candesartan alone at a high concentration lowered blood pressure in SHR rats and that CGP42112A only provided a depressant effect in the presence of candesartan [182].”
Would it be better to eliminate the section Acknowledgments? Seems to be not edited from a template
This section has been removed.
References include the month of the year in French?
References have been homogenized.
Round 2
Reviewer 2 Report
The authors adequately addressed all the concerns raised by this reviewer, even those related to comments made by this reviewer regarding the introductory paragraph, that were based on my ignorance about the topic of this series, and for this I apologize.
In my opinion, no more changes are required for publication